# Epithelial–Mesenchymal Transition in Liver Fluke-Induced Cholangiocarcinoma

**DOI:** 10.3390/cancers13040791

**Published:** 2021-02-14

**Authors:** Kanlayanee Sawanyawisuth, Goro Sashida, Guojun Sheng

**Affiliations:** 1Department of Biochemistry, Faculty of Medicine, Khon Kaen University, Khon Kaen 40002, Thailand; 2Laboratory of Transcriptional Regulation in Leukemogenesis, International Research Centre for Medical Sciences, Kumamoto University, Kumamoto 860-0811, Japan; 3Laboratory of Developmental Morphogenesis, International Research Centre for Medical Sciences, Kumamoto University, Kumamoto 860-0811, Japan

**Keywords:** cholangiocyte, cholangiocarcinoma, bile duct, liver fluke, epithelial–mesenchymal transition

## Abstract

**Simple Summary:**

Parasitic infection remains a health threat in many countries. Liver flukes, parasitic flatworms endemic to southeast and east Asia, cause bile duct inflammation and are major risk factors of bile duct cancer (cholangiocarcinoma). As the only group of eukaryotic organisms listed as carcinogens, liver flukes can increase cholangiocarcinoma incidence by 100-fold in some parts of Thailand. How they interact with bile duct epithelial cells during tumor initiation and progression is unknown. In this review, we summarize molecular and cellular evidence linking liver fluke-associated cholangiocarcinoma with mis-regulation of epithelial–mesenchymal transition (EMT), a multicellular morphogenetic process known to be involved in many normal and pathological settings, including cancer. EMT markers and regulators can potentially be used to facilitate cholangiocarcinoma diagnosis and treatment.

**Abstract:**

Cholangiocarcinoma (CCA) is the second most common type of hepatic cancer. In east and southeast Asia, intrahepatic CCA is caused predominantly by infection of *Opisthorchis viverrini* and *Clonorchis sinensis*, two species of parasitic liver flukes. In this review, we present molecular evidence that liver fluke-associated CCAs have enhanced features of epithelial–mesenchymal transition (EMT) in bile duct epithelial cells (cholangiocytes) and that some of those features are associated with mis-regulation at the epigenetic level. We hypothesize that both direct and indirect mechanisms underlie parasitic infection-induced EMT in CCA.

## 1. What Is Cholangiocarcinoma?

Cholangiocarcinoma (CCA) is a cancer of bile duct epithelial cells (cholangiocytes). It is the second most common type (15%) of liver cancers worldwide, after hepatocellular carcinoma (HCC). CCAs account for about 3% of all gastrointestinal tract cancer and, depending on their origins in the biliary tree, are divided into intrahepatic (iCCA), perihilar (pCCA), and distal (dCCA) CCAs [1] (Figure 1A). iCCAs represent a small percentage (10%) of all CCAs [1,2]. However, in some regions of the world (e.g., northeast of Thailand), CCA incidence (mainly of the iCCA type) could be 100 times higher than the global average, and is the main cause of mortality of all cancer types combined [3,4] (World Health Organization’s Global Cancer Observatory). Increased CCA risk in Thailand has been postulated to be associated with consumption of raw or undercooked fish infected with liver fluke *Opisthorchis viverrini* [3,5,6,7,8], which excysts (leaving its protective cyst) in the duodenum after being ingested by humans, and then matures into adulthood in the bile duct.

## 2. What Are Flukes

*Opisthorchis viverrini* (*O. viverrini*) is one of several parasitic platyhelminths (flatworms) which use humans as a host in their life cycle (Figure 1B). It is endemic to southeast Asia and belongs to a class of flatworms called *Trematoda* (flukes). After infecting humans, *O. viverrini* and *Clonorchis sinensis,* another fluke species endemic to China, take residence in the biliary duct (hence the name liver fluke). Together, these two species affect 25–40 million people (depending on estimates) in east and southeast Asia [3,6,8,9], causing acute and chronic biliary inflammation and CCAs. Another Trematode genus, *Schistosoma* (also known as urinary and intestine blood flukes), is the most prevalent parasitic group for humans, affecting approximately 240 million people worldwide and causing schistosomiasis and bladder cancer (World Health Organization’s fact sheet on Schistosomiasis). *O. viverrini*, *C. sinensis*, *and Schistosoma haematobium* are the only known eukaryotic organisms with strong evidence for cancer association and have been classified as group 1 carcinogens (biological agents that cause cancers) [10]. Other group 1 carcinogens include bacterium *Helicobacter pylori* and viruses like HIV-1, HTLV-1, human papillomavirus (HPVs), HBV, HCV, and EBV [10].

An adult liver fluke (*O. viverrini*) is a flat-shaped worm about 7 mm long and 1.5 mm wide [11]. It reaches sexual maturity one month after metacercaria, its larval form, enters the biliary system through ampulla of Vater, the common opening of the bile and pancreatic ducts into the duodenum (Figure 1A, Figure 2A). Adult *O. viverrini* is a cross-fertilizing hermaphrodite (each worm having both male and female reproductive organs) and can live for up to two decades in infected human individuals. It attaches to the bile duct epithelium through its oral and ventral suckers and feeds on bile fluid, cholangiocyte cell fragments, and blood through its blind-ended endoderm-derived digestive system and its ectoderm-derived syncytial neodermis. In regions of Thailand with high prevalence of *O. viverrini,* fecal examination shows infection in up to 80% of human adults. Most cases of *O. viverrini* infection are asymptomatic. A small percentage of cases develop into chronic inflammation and CCAs (with the highest incidence of 85 cases per 100,000 population in Northeast Thailand) [4].

## 3. Flukes and Cholangiocarcinoma

Although association between liver fluke infection and CCA has been well documented, the exact nature of CCA tumorigenicity is less clear. An exome sequencing-based analysis [5] of *O. viverrini*-associated CCAs revealed 206 somatic mutations in 187 genes as potential CCA-associated genetic alternations, most of which (about 90%) being missense or nonsense mutations. They further validated 15 of those genes in additional CCA samples, and the top three of those mutations were in the *TP53* (44%), *KRAS* (17%), and *SMAD4* (17%) genes. All three genes have been associated with other types of tumors with similar frequencies [12], suggesting that they may not be specific signatures of fluke-induced CCAs. Additional mutations were seen in the *MLL3*, *ROBO2*, *RNF43*, *PEG3*, and *GNAS* genes, some of which having been reported as epigenetic modifiers or cellular morphogenetic regulators [13,14,15,16], suggesting that both genetic- and epigenetic-level changes in cholangiocytes contribute to CCA progression.

In a recent study of genomic and epigenetic analyses of 489 patients with CCA [17], fluke-positive CCA specimens showed significant enrichment in genetic mutations of the *TP53*, *ARID1A*, and *BRCA1/2* genes, in contrast to fluke-negative CCA showing mutations in the *IDH1/2* and *BAP1* epigenetic modifier genes, suggesting a bias towards selection of specific cancer mutations during fluke-associated CCA progression. Promoter hyper-methylation silences gene expression and has been implicated in tumor initiation and malignant transformation. In fact, in comparison with fluke-negative CCAs, fluke-positive CCAs showed DNA hyper-methylation in polycomb repressive complex 2 (PRC2)-target promoter CpG islands, accompanied by a higher incidence of C-to-T mutation caused by deamination of methylated cytosine. Cancer cells have been shown to promote DNA hyper-methylation in promoter regions of bivalent genes marked by MLL-mediated H3K4me3 and PRC2-mediated H3K27me3, which are known to be activated upon the differentiation of ES cells [18,19], suggesting that PRC2 fine-tunes the level of promoter DNA methylation in normal cells, and that functional impairment of PRC2 may promote DNA hyper-methylation and subsequent methylation-mediated mutations to drive the development of cancer.

In addition to those cholangiocyte-associated alterations during the course of chronic infection, exogenous factors are also assumed to be at play. Those factors include duct epithelial tissue damage and repair due to liver fluke infection, host immunopathological responses to chronic infection, and direct secretory stimulation from the parasite (Figure 3A).

## 4. Cholangiocarcinoma and EMT

Epithelial–mesenchymal transition (EMT) and its reverse process (MET) are dynamic morphogenetic events leading to changes in multicellular-level organization [20]. Collectively referred to as EMTs, they are commonly seen in animal development and are essential for tissue and organ formation [21,22]. EMTs have been increasingly implicated in pathological processes of tissue malformation, including in cancer progression and tissue fibrosis [20,22,23]. The bile duct is an endoderm-derived epithelial structure (Figure 2A,B). Like other endoderm-derived epithelia (e.g., esophageal, lung, gastric, hepatocellular, colorectal, and prostate), the bile duct (cholangiocyte) epithelium is prone to cancer formation. EMT’s involvement in cancer is multifold, including the process of breaching the primary epithelial organization during tumor initiation, creating cellular and microenvironment heterogeneities of primary tumors and modulating metastatic potentials of primary tumor cells during intravasation and extravasation [20].

The involvement of EMT-like processes in CCAs has been proposed previously [1,24,25], supported by evidence of disruption in epithelial cell–cell junctions and promotion of mesenchymal characteristics. These data (summarized in the above-mentioned papers) (Table 1; left column) include altered distribution and expression of E-cadherin (an epithelial adherens junction marker), N-cadherin (a mesenchymal adherens junction marker), β-catenin (an intracellular mediator of cadherin-based cell–cell interaction and a regulator of WNT signaling), Cytokeratin 19 (CK19, an epithelial-type intermediate filament marker), vimentin (a mesenchymal-type intermediate filament marker), S100A4 (a regulator of metalloproteases and of E-cadherin), and EMT-associated transcription factors Snail (Snai1), Slug (Snai2), Twist (Twist1), Zeb1, and Zeb2. In addition, cytokine signaling commonly associated with EMT regulation has also been shown to participate in CCA progression, including TGFβ1, 2, and 3 and TGFβ receptors 1 and 2 (expressed in both CCA tumor cells and surrounding stroma cells), chromatic regulator Hmgb1 (a positive regulator of TGFβ-induced EMT), BMP7 (a negative regulator of TGFβ-induced EMT), TNFα (a positive regulator of E-cadherin and CK19 and negative regulator of S100A4, Snail, and Zeb2), IL-6 (expressed by CCA tumor cells and surrounding cells, and inhibiting epithelial and promoting mesenchymal characteristics), SOCS3 (inhibitor of IL-6 induced EMT), SMAD4 (converging point of IL-6- and TGFβ1-induced EMT), EGFR (promoting cytoplasmic localization of E-cadherin), EphA2 (inhibiting epithelial junctions and promoting mesenchymal markers), H4HR (inhibiting fibronectin, vimentin, and S100A4 expression and promoting CK7, CK8, and CK19 expression), CXCR4 (promoting Slug, vimentin, and MMP-9 expression and cell migration), Notch1 (upregulating Snail, Sox9, αSMA, and vimentin levels and downregulating E-cadherin levels), Hedgehog signaling (downregulating E-cadherin), and several EMT-related microRNAs (miR-221, miR-200c, miR-204, miR-214, miR-34a, and miR-21).

The involvement of glycosylation (a type of protein posttranslational modification) in EMT-dependent migration and invasion of CCA was recently unveiled. O-GlcNAcylation, the addition or removal of N-acetylglucosamine (GlcNAc) onto glycoprotein, is reversibly controlled by O-GlcNAc transferase (OGT) and N-acetyl D-glucosaminidase (OGA). Suppression of OGT by siRNA (siOGT) effectively reduced cell migration and invasion of CCA cells whereas siOGA showed opposite effects. Manipulation of O-GlcNAcylation levels affected the nuclear translocation of NF-κB and phosphorylation of Akt, together with expression of matrix-metalloproteinases (MMPs) including MMP2, MMP7, and MMP9 [37]. High glucose levels also increased O-GlcNAcylated vimentin, which led to an increase in its stability and promoted CCA cell motility [38]. Heterogeneous nuclear ribonucleoprotein-K (hnRNP-K) was discovered as a novel O-GlcNAcylated protein related to the progression of CCA cells. Diminution of hnRNP-K expression by siRNA markedly decreased the motility of CCA cells via increases in E-cadherin and claudin-1 expression and decreases in vimentin and Slug expression. In addition, suppression of hnRNP-K diminished the expression of MMP2 and MMP7 [39].

## 5. Fluke-Associated Cholangiocarcinoma and EMT

These lines of evidence, mentioned above, strongly support the involvement of EMT in CCA progression. These data, however, have been obtained from CCA tumor samples or cell lines that are not associated with liver fluke infection. It is therefore unclear whether the association observed between liver fluke infection and CCA incidence can be explained by enhanced molecular and cellular features that promote cholangiocyte EMT. Table 1 (right column) summarizes data from published studies in support of a link between EMT and fluke-associated CCAs. In the aforementioned exome sequencing of liver fluke-associated CCAs [5]; four of the genetic mutations uncovered in that analysis are known to be associated with EMT regulation (SMAD4, MLL3, ROBO2, and RNF43). Two separate studies using the hamster model of fluke-induced CCAs showed high levels of expression of terminal fucose in CCAs and that inhibition of fucosyltransferase 1 activity in liver fluke-associated CCA cell lines reduced mesenchymal marker (Slug, vimentin, and S100A4) expression and increased epithelial marker (E-cadherin and Claudin1) expression [26]. The expression of O-GalNAcylated glycans was elevated in liver fluke associated with CCA tissues from hamsters. GalNAc transferase 5 (GALNT5) was a major enzyme in liver fluke-associated human CCA cell lines with high O-GalNAcylated glycan. Knockdown of GALNT5 expression inhibited migration and invasion of CCA cell lines via an increase in claudin-1 and a decrease in slug and vimentin expression [27]. A study on TNFα’s effort on EMT markers Zeb2 and S100A4 supported involvement of EMT regulation in liver fluke-associated CCA [28]. It has also been shown that anti-parasitic drugs (Xanthohumol and Praziquantel) could alleviate inflammation and reduce expression levels of EMT transcription factor Twist [29]. In another study, a stress-response gene, Gadd45b, was shown to be highly expressed in clinical CCA samples and that in a liver fluke-associated CCA cell line, Gadd45b knockdown led to a reduction in mesenchymal and an increase in epithelial marker expression [30]. Interestingly, two liver-fluke associated CCA cell lines with differential epithelial and mesenchymal characteristics also exhibited differential metastatic capacities in animal models [31]. This is in agreement with the emerging consensus on cancer EMT, supported by data from many cancer models, pointing an association between a tumor cell’s partial EMT features and its metastatic malignancy [40,41]. Finally, consistent with these findings, *O. viverrini* is known to secrete granulin [32,33] and glutathione S-transferase [35], both known as potent mitogens and EMT inducers [34,35,36,42]. Taken together, these data suggest that one of the main features in liver fluke-associated CCAs is to alter epithelial and/or mesenchymal cell characteristics of cholangiocytes, and that increased incidence of CCAs in fluke-endemic regions of the world is primarily the consequence of increased levels of EMT stimulation, from either local tissue environment, the host immune system, or parasitic secretory/excretory products (Figure 3A,B).

## 6. Outlook

Cancer arises through genetic and/or epigenetic-based malfunction of normal tissue homeostasis, including failure in controlling cellular proliferation and differentiation, and in maintaining tissue-level cell-cell and cell-matrix organizations. The former is often associated with, and is likely also a prerequisite of, primary tumor initiation. Its molecular markers are involved in cell-cycle control, manifested at multiple regulatory levels, from receptors to epigenetic modifiers. The latter is associated with both the initiation and metastasis of primary tumors and can be described as tumor cells’ niche structure within the context of surrounding host tissues. This involves mostly tumor (or tumor-prone) cell-cell and cell-matrix contacts and tumor-host interactions (e.g., with the host’s vascular and immune systems). Pathogenetic triggers in this respect are likely twofold. On one hand, fluke infection-induced innate and adaptive immune responses, although not the focus of this review, are known to increase risks of CCA progression through immunomodulation of tumor microenvironment and of chronic bile duct inflammation which secondarily promotes CCA [3,43,44,45]. On the other hand, epithelial and mesenchymal characteristics of tumor cells have been shown in many experimental models to be an important predictor of tumor malignancy. As discussed in this review, we support the notion that mis-regulation of EMT characteristics is a major driver in CCA progression.

Genetic and epigenetic alternations leading to cancer malignancy can be categorized as either spontaneous or induced. Spontaneous alternations are the consequence of normal cellular history and are associated epigenetically with progressive restriction of cellular differentiation and proliferation potentials and genetically with accumulation of errors during normal DNA replication and repair. Induced alternations are due to physical (e.g., irradiation), chemical (e.g., tobacco smoking), and biological carcinogens (e.g., HPV infection), and their associated host responses (e.g., chronic inflammation), the latter two of which are likely the main contributors to fluke-associated CCAs. Although the major risk factor of CCA in western countries is primary sclerosing cholangitis, an inflammatory disease not related to parasitic infection and different from fluke-associated cholangitis [46,47], more studies will be needed to probe whether host responses to chronic inflammation may constitute a shared underlying mechanism for all CCAs. Furthermore, it is important to point out that currently there are only a very small number of biological agents (viruses, bacteria, and eukaryotic organisms) that have been clearly categorized as carcinogens, despite the vast array of other human diseases that are known to be associated with their infection. This suggests that mechanistically, the overlap between human cancer and human viral, bacterial, and parasitic infection is surprisingly small. When they do overlap, however, the levels of association are relatively strong. For instance, almost all cervical cancer cases can be associated with human papillomavirus (HPV) infection; *H. pylori* infection is the number one risk factor for gastric cancer; and as discussed in this review, *O. viverrini* infection can result in up to 100x increase in CCA incidence.

We laid out a scenario in which EMT regulation lies at the crossroads of liver fluke infection and CCA progression. This is linked to yet uncharacterized, specific EMT-promoting properties of parasitic liver fluke on human bile duct epithelium and is partially mediated through chronic inflammation of the biliary tree. Finding EMT-based diagnostic and therapeutic targets of CCAs will likely require collaboration between researchers working in the parasitology, epidemiology, cancer biology, and EMT research fields.

## 7. Conclusions

EMT is involved in liver fluke-associated CCAs. Finding appropriate EMT-related markers may facilitate early diagnosis of CCAs in liver fluke-infected populations.

## Figures and Tables

**Figure 1 cancers-13-00791-f001:**
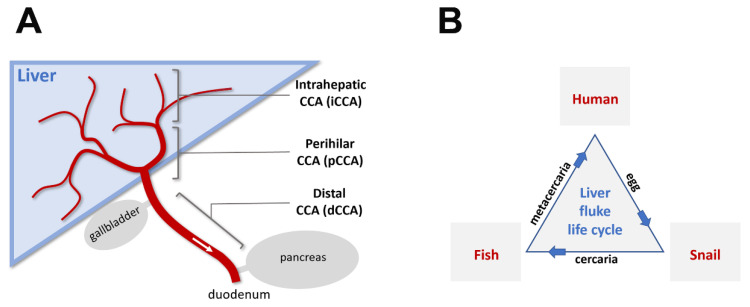
Schematic diagram of cholangiocarcinoma (CCA) types and the liver fluke life cycle. (**A**): The bile duct and its relationship to the liver, gallbladder, pancreas, and duodenum. The biliary tree (red) is composed of the ampulla of Vater (opening to the duodenum part of the small intestine after joining of the pancreatic duct), common bile duct (after joining of the cystic duct from gallbladder, before joining of the pancreatic duct), common hepatic duct (after joining of the left and right hepatic ducts, before joining of the cystic duct), left and right hepatic ducts, and progressively narrower intrahepatic ducts and ductules. CCAs are categorized based on the origin of tumorigenic cholangiocytes within the biliary tree. Fluke-induced CCAs are primarily of the intrahepatic type, likely due to comparable lumen diameters with respect to the size of adult parasitic flukes. (**B**): Life cycle of liver flukes in humans. Parasitic liver flukes (*Opisthorchis viverrini* and *Clonorchis sinensis*) use the human bile duct as a host environment for sexual maturation. Eggs generated from cross-fertilization of hermaphroditic adult liver flukes pass through feces and infect the first intermediate host (snails) after a brief period of free-living development. Asexual reproduction takes place in this host environment. Larval flukes (cercariae) released from infected snails metamorphose into encysted metacercariae either in a natural environment or in a second intermediate host (freshwater fish for liver flukes in humans). The encystment in fish muscular tissues presumably confers evolutionary advantage for liver flukes in reaching their final parasitic host environment, where they can live for up to two decades.

**Figure 2 cancers-13-00791-f002:**
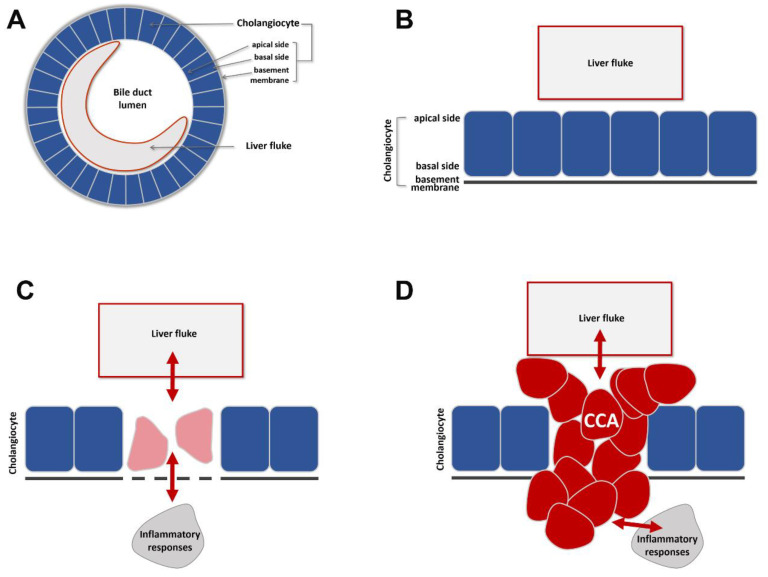
Schematic diagram of how fluke infection in the bile duct leads to CCA. (**A**,**B**): An adult liver fluke resides within the lumen of a comparably sized bile duct. Bile duct epithelial cells are called cholangiocytes. Their apical surface faces the lumen and their basal surface is covered by a supporting, non-cellular basement membrane. (**C**): Fluke infection induces damage, repair, and other genetic and epigenetic changes in surrounding cholangiocytes. This may be mediated indirectly through host responses to infection. (**D**): CCAs arise as a consequence of chronic fluke infection.

**Figure 3 cancers-13-00791-f003:**
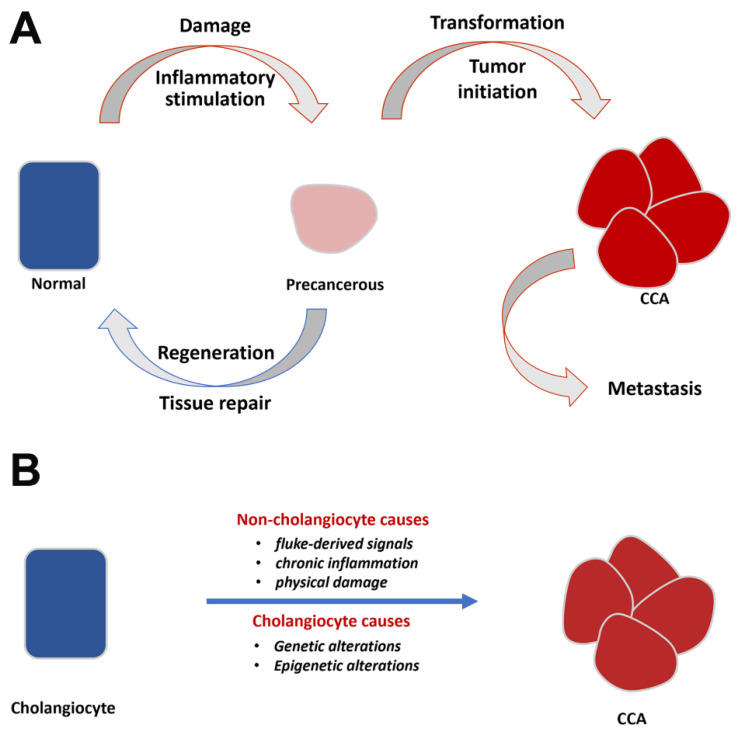
Putative epithelial–mesenchymal transition (EMT)-related factors that influence CCA progression and summary of CCA’s biological causes. (**A**): Putative EMT-related factors. Fluke-derived physical and biochemical stimulation and host inflammatory responses lead to repeated cycles of tissue damage, repair, and regeneration and to cholangiocyte morphological changes during which EMT regulation may play an importance role. In the course of chronic fluke infection, tumor initiation leads to a breach of the cholangiocyte basement membrane and local and distal metastases, all of which involve modulation of epithelial/mesenchymal characteristics of cholangiocarcinoma cells. (**B**): Summary of biological causes leading to transformation from cholangiocyte to CCA. Cholangiocyte-related causes include genetic and epigenetic alterations either occurring spontaneously or as a consequence of fluke infection. Non-cholangiocyte-related causes include both parasite-induced and host inflammatory response-induced changes to cholangiocytes and their extracellular environment.

**Table 1 cancers-13-00791-t001:** Molecular and cellular evidence linking EMT with CCA. Left column: EMT and CCA; Right column: EMT and fluke-associated CCA.

Molecular/Cellular Evidence Linking EMT and CCA (Based on Banales et al. [1], 2020; Nitta et al [25]., 2014; Vaquero et al. [24], 2017)	Molecular/Cellular Evidence Linking EMT and Fluke-Associated CCA
TGFb1,2,3; TGFb receptor 1,2; SMAD4	SMAD4; MLL3; ROBO2; RNF43 (Ong et al., 2012 [5])
HMGB1	FUT1 (Fucosyltransferase 1) (Indramanee et al., 2019 [26])
BMPR; EGFR; CXCR; EPH; H4HR; CXCR4; NOTCH1; HH pathways	GALNT5 (GalNAc transferase 5) (Detarya et al., 2020 [27])
TNFa	TNFa (Techasen et al., 2014 [28])
IL-6; SOCS3	Anti-parasitic drugs Xanthohumol and Praziquantel (Kimawaha et al., 2020 [29])
microRNAs (miR-221, miR-200c, miR204, miR214, miR34a, miR21)	GADD45B (Myint et al., 2018 [30])
Altered expression/distribution of E-cad, N-cad, b-catenin, Cytokeratin 19, vimentin, and S100A4	Fluke-associated CCA cell lines with different E/M characteristics show different metastatic capacities (Saensa-Ard et al., 2017 [31])
Altered expression/distribution of SNAI1, SNAI2, TWIST1, ZEB1, and ZEB2	Fluke-secreted granulin and GST (Arunsan et al., 2020 [32]; Smout et al., 2009 [33]; Tanimoto et al. [34], 2016; Daorueang et al. [35], 2012; Arechavaleta-Velascro et al., 2017 [36])

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
