# Peer review of "Epithelial–Mesenchymal Transition in Liver Fluke-Induced Cholangiocarcinoma"

_cancers, 2021, doi:10.3390/cancers13040791_

Round 1

Reviewer 1 Report

This is an informative review on “liver fluke induced cholangiocarcinoma”.

General comment: the review could give more details in terms of (alternative) mechanistic/morphological principles, clinical relevance (diagnosis, prognosis) and role of underlying disease in the different mechanisms of EMT pathways (comparison liver fluke induced CCA versus other causes of CCA).

The second part of the manuscript is difficult to read, particularly regarding the differences in EMT pathways between liver fluke induced CCA and other types/causes of CCA’s.

The authors state in their ‘Outlook’ line 257-259 “This is linked to yet uncharacterized, specific EMT promoting properties of parasitic liver fluke on human bile duct epithelium and is partially mediated through chronic inflammation of the biliary tree.”

PSC is also an inflammatory disease associated with increased risk of iCCA in Western countries. What are the major and most relevant differences, despite inflammatory background in both disease?  

It would improve the manuscript if the specificity of EMT pathways in fluke induced CCA is more clearly/systematically described, in comparison to shared pathways that are seen in CCA/tumors in general.

Paragraph 4 addresses EMT in CCA in general; in paragraph 5 EMT in fluked induced CCA is described. It is difficult to follow in the text which of these pathways are shared pathways for CCA in general and which pathways are ‘specific’ for fluke induced CCA.

In Paragraph 3, it is described that genes, such as KRAS, TP53 en SMAD4 are seen in different tumor types, including fluke induced CCA. Could it be that the differences in IDH1/2 expression that the authors report subsequently in the text (contributed to fluke negative CCA), depend on iCCA subtype (small and large duct types) and not so much on the cause of disease (liver fluke vs other causes of CCA)? 

The authors explain in the introduction the different anatomical subtypes of CCA. It is not clear for some of the summarized studies (on CCA in general), if these are addressing pathways in iCCA or also pCCA and/or dCCA. It maybe helpful to provide a more detailed description on EMT in the 3 anatomical subtypes and discuss the (ir)relevancy of this in the comparison with fluke induced CCA. NB. In the context of obstruction due to parasites of the smaller bile ducts, ductular reaction/fibrosis may occur, which may interfere with EMT signals.

Almost by definition, malignant transformation and invasive growth of epithelial cells involves ‘epithelial mesenchymal transition’ So why is EMT of specific relevance in Liver Fluke-Induced Cholangiocarcinoma and what is the clinical relevance in terms of diagnosis, prognosis etc?

Could variability in EMT contribute to differences in outcome between Eastern and Western patients as described for pCCA (PMID: 30087051)? Or, like in other cancer types, such as colon cancer, in which tumor budding, as a manifestation of EMT, is associated with worse prognosis? Or does this need more collaborative studies? 

Minor: Check spelling: Figure 3. A schematic diagram of how fluke infection in the bike duct leads to CCA

Author Response

our responses to  reviewer 1 comments/questions are included in the pdf file.

Reviewer 2 Report

  • The legend to Fig 4 reads, " Putative EMT related factors that influence CCA progression". However, no such EMT related factors are explicitly illustrated in the actual figure. This figure merely resembles a general schema for the process of carcinogenesis. Unless the involvement of fluke induced tissue damage and biochemical stimulation in the inflammatory response that may mediate CCA development is specifically highlighted in the figure, this figure remains redundant. 
  • The various EMT markers, signaling proteins associated with EMT regulation, and EMT-related microRNAs whose expression has been reported to be altered in CCA (described from line 147-168) may be enlisted in a table along with their functions. This shall provide a concise overview of all such known EMT markers and related factors that are linked to CCA.
  • Headings of sections 1 and 2 should have a '?' (What is cholangiocarcinoma?)
  • Line 58, replace 'Others' with 'Other'
  • Line 85, replace 'bike' with 'bile'
  • Line 86, 'comparably-size' should be ' comparably-sized'
  • Line 109, correct "in during"
  • Line 195. change 'increase' to 'increased'
  • Line 223, change 'fluke-injection' to 'fluke-infection'

Author Response

our responses to reviewer 2 comments/questions are included in the pdf file.

Reviewer 3 Report

This is a very good review on how the infection of two species of parasitic liver flukes promotes the development of cholangiocarcinoma (CCA). I suggest that the authors can review more on the innate and adaptive immunity being triggered after the infection of parasitic liver flukes. It is because it is well-known the chronic infection and cancer-related inflammation are important factors in cancer promotion and development.

Author Response

our responses to reviewer 3 comments/questions are included in the pdf file.

Round 2

Reviewer 1 Report

I thank the authors for the revised manuscript. The changes made are a clear improvement.